# Application of Machine Learning to Predict CO_2_ Emissions in Light-Duty Vehicles

**DOI:** 10.3390/s24248219

**Published:** 2024-12-23

**Authors:** Jeffrey Udoh, Joan Lu, Qiang Xu

**Affiliations:** Department of Computer Science, School of Computing and Engineering, University of Huddersfield, Queensgate, Huddersfield HD1 3DH, UK; u2186837@unimail.hud.ac.uk (J.U.); j.lu@hud.ac.uk (J.L.)

**Keywords:** sensors, machine learning, CO_2_ emission, transport, regression analysis

## Abstract

Climate change caused by greenhouse gas (GHG) emissions is an escalating global issue, with the transportation sector being a significant contributor, accounting for approximately a quarter of all energy-related GHG emissions. In the transportation sector, vehicle emissions testing is a key part of ensuring compliance with environmental regulations. The Vehicle Certification Agency (VCA) of the UK plays a pivotal role in certifying vehicles for compliance with emissions and safety standards. One of the primary methods employed by the VCA to measure vehicle emissions for light-duty vehicles is the Worldwide Harmonized Light Vehicles Test Procedure (WLTP). The WLTP is a global standard for testing vehicle emissions and fuel consumption, and sensors are crucial in ensuring accurate, real-time data collection in laboratories. Using the data collected by the VCA, regression machine learning models were trained to predict CO_2_ emissions in light-duty vehicles. Among six regression models tested, the Decision Tree Regression model achieved the highest accuracy, with a Mean Absolute Error (MAE) of 2.20 and a Mean Absolute Percentage Error (MAPE) of 1.69%. It was then deployed as a web application that provides users with accurate CO_2_ emission estimates for vehicles, enabling informed decisions to reduce GHG emissions. This research demonstrates the efficacy of machine learning and AI-driven approaches in fostering sustainability within the transportation sector.

## 1. Introduction

Climate change is one of the most pressing global challenges, characterized by rising temperatures, extreme weather events, and significant disruptions to ecosystems and human society [1]. Consequently, reducing greenhouse gas emissions, especially of CO_2_, has become a shared and essential objective to mitigate the effects of climate change [2]. The report by BEIS (Department for Business, Energy & Industry Strategy, UK, now replaced by Department for Energy Security and Net Zero, Department for Science, Innovation and Technology, and Department for Business and Trade, UK) revealed that the transport sector had the highest greenhouse gas emissions in 2021, and made up 26% of the total emitted greenhouse gases [3].

The rapid advancement of the global technological revolution has sped up the digital transformation of numerous industries, and climate change has been significantly impacted by digital transformation across various sectors [4]. Digital transformation in transportation refers to the integration of electronic data and cutting-edge technology that provides an efficient and effective service delivery in the sector [5]. Various integrations of data and information technologies, including but not limited to Internet of Things (IoT) devices, Artificial Intelligence (AI), big data, and smart infrastructure, aim to improve efficiency, implement effective pricing systems, enhance safety for users, and promote sustainability within the transportation sector [6].

Vehicle emissions testing is a key part of ensuring compliance with environmental regulations. In vehicle emissions testing, the data collection process is critical for determining the amount and type of pollutants emitted by vehicles under simulated real-world driving conditions. In the WLTP, various sensors are used to ensure that data are captured accurately and in real time. These sensors are integrated into different stages of the testing process, including the measurement of exhaust gases, particulate matter, ambient conditions, and vehicle performance on the chassis dynamometer [7].

### 1.1. Research Objectives

The objectives and contribution of this research are as follows:i.An exploration of the ideas and conceptual framework of digital transformation in transportation, and the impact on GHG emissions.ii.The identification of significant features influencing vehicular CO_2_ emissions.iii.The development and evaluation of an ML model for predicting CO_2_ emissions in light-duty vehicles.iv.The deployment of a web-based application as a practical tool for stakeholders to estimate the CO_2_ emitted by a vehicle.

### 1.2. Problem Statement

Since the 1980s, there has been growing concern about climate change due to human activities that emit greenhouse gases. The United Nations Environment Programme (UNEP) and the World Meteorological Organization (WMO) established the Intergovernmental Panel on Climate Change (IPCC) to oversee climate change and its related activities. Since its inception, the IPPC has published five comprehensive reports, and these reports all concluded that the Earth is warming, and this is caused by greenhouse gas emissions from human activities [8]. A report by the Emissions Database for Global Atmospheric Research (EDGAR) revealed that over the years, there has been an increase in global greenhouse gases as a result of the increase in CO_2_ emissions from various countries, and countries like China, the United States, the EU27, India, Russia, and Japan are considered the biggest emitters of CO_2_ and other greenhouse gases [9]. A report by the U.S. Environmental Protection Agency reveals that carbon dioxide made up 65% of the global greenhouse gases emitted [10], while a report by the Department for Business, Energy & Industrial Strategy released in 2023 revealed that, in 2022, CO_2_ made up 80% of the greenhouse gas emitted in the UK [3]. See Figure 1.

The transportation sector is made up of industries that provide services to enable the movement of people and goods from one point to another; this includes airlines, roads, rail, air freight, sea freight, etc. [11]. It is a significant contributor to air pollution in both urban and rural areas due to its heavy reliance on fossil fuels [12]. Over the years, there has been a steady increase in the amount of greenhouse gases emitted by the industries in the transport sector, having an annual average growth rate of 1.7% from the year 1990 to 2022, surpassing the annual average rate of the other end-use sectors [13]. 

### 1.3. Research Scope

The scope of this research includes a discussion on the digital transformation of the transportation sector, and the predictive analysis of CO_2_ emitted by passenger vehicles, using the data collected by sensors. The machine learning predictive model will consider the important features that correlate with CO_2_ emissions in light-duty vehicles, and use them in developing a predictive model with high accuracy. The case study of this research is on the United Kingdom, as the United Kingdom is considered as a leading advocate for climate change awareness.

## 2. Literature Review

### 2.1. The Concept of Digital Transformation

Digital transformation is understood to be a fundamental change in the utilization of digital technologies to improve efficiency, enrichment, and innovation. Digital transformation is a comprehensive initiative that integrates digital technologies into every function of an organization, transforming its delivery model and value proposition [14]. It goes beyond simply acquiring digital tools; it involves drastic change and reworking business structures to suit the transformation of needs in a digitized world [15].

Also, digital transformation can refer to the introduction of advanced technologies including Artificial Intelligence (AI), machine learning (ML), and Internet of Things, and data analysis capabilities. These technologies make up the digital spine that allows companies to gather, process, and gain insight from mountains of data. Delivering improved and innovative experiences is one of the main drivers of digital transformation. Digital tools are used by organizations to identify customer preferences, customize engagement, and provide smooth experiences [14]. However, data are at the heart of digital transformation since analytics allows for the extraction of actionable insights. The aim is to transform raw data into useful information that guides decision-making practices, reveals patterns, and helps introduce a more reactive approach toward changes in market dynamics. This data-informed approach is also applicable to customer behavior, preferences, and needs that can be used in organizations where products or services need adjustment according to evolving requirements.

In addition, digital transformation is about improving or re-engineering the business processes already in place. This entails a detailed assessment and remodeling of the workflows so that they suit what is available in digital technologies. Organizations use data analytics and AI to forecast equipment failure and perform preventative maintenance ahead of time, thus minimizing inactivity [16].

### 2.2. Emerging Technologies and Their Contribution to Transformation in Transportation

The transportation sector has several emerging technologies, some fully functional, providing quality data and information to tackle issues, while others are still yet to achieve a wider audience to obtain adequate data for use. The vast number of emerging technologies include the Internet of Things (IoT), which is widely used for inventory and fleet management, payment, and ticketing; this provides data that can be analyzed and interpreted. Artificial Intelligence, although still contributing to transportation transformation, within the next decade, is expected to provide quality data for use. These emerging technologies will improve the efficiency of the transportation system, with exponential increments expected within the next decade.

#### 2.2.1. Internet of Things (IoT) in Transportation

The Internet of Things (IoT) has emerged as a key enabler in the transportation sector. IoT involves connecting physical devices and vehicles to the internet, allowing them to collect and exchange data in real time. In transportation, it refers to the interconnected network of physical devices and vehicles embedded with sensors, software, and other technologies to collect and exchange data. This interconnected ecosystem facilitates the seamless flow of information, enabling smarter and more efficient transportation systems. Through IoT, vehicles become intelligent nodes in a broader network. Sensors embedded in vehicles monitor performance metrics, fuel consumption, and maintenance needs. This real-time data allow for predictive maintenance, reducing downtime and enhancing the lifespan of transportation assets [17].

One crucial aspect of IoT in transportation is in the domain of Intelligent Transportation Systems (ITSs). These systems leverage IoT technologies to optimize traffic flow, vehicle congestion, and overall safety on roads. For example, connected vehicles can communicate with each other and with traffic infrastructure to receive real-time updates on road conditions, accidents, and other critical information [18]. This enables drivers to make informed decisions, improving road safety and reducing the likelihood of accidents.

IoT also plays a pivotal role in fleet management as companies can utilize IoT devices to monitor the condition and performance of their vehicles in real time. Sensors on trucks or buses provide data on fuel efficiency, engine health, and maintenance needs. This proactive approach to vehicle management helps reduce downtime, lowers maintenance costs, and contributes to the overall sustainability of the transportation fleet [19].

The Internet of Things is reshaping the landscape of transportation by fostering connectivity, automation, and data-driven decision-making. The integration of IoT technologies in vehicles, infrastructure, and logistics systems enhances safety, efficiency, and sustainability, ultimately contributing to the evolution of smarter and more resilient transportation ecosystems. Smart traffic management and route optimization both contribute to reduced fuel consumption and emissions. In addition, IoT sensors can monitor the air quality and provide insights for urban planners to implement sustainable transportation policies. IoT facilitates smart traffic management, enabling cities to optimize traffic flow and reduce congestion through real-time monitoring and data analysis [20].

#### 2.2.2. Artificial Intelligence (AI) in Transportation

Artificial Intelligence (AI) is revolutionizing the transportation sector by introducing innovative solutions that enhance efficiency, safety, and sustainability. In the realm of transportation, AI refers to the application of advanced algorithms and computational models to process, analyze, and interpret data, enabling intelligent decision-making across various components of the transportation ecosystem. In autonomous vehicles, AI algorithms analyze data from sensors, cameras, and radar systems to make real-time driving decisions. This technology holds the promise of safer and more efficient transportation, with the potential to reduce accidents and improve traffic flow [21].

One significant application of AI in transportation is in the development of autonomous vehicles. These vehicles leverage machine learning algorithms and sensor technologies to perceive their environment, interpret road conditions, and make decisions without human intervention. Companies such as Waymo and Tesla have been at the forefront of integrating AI into self-driving cars, aiming to reduce accidents, increase road safety, and optimize traffic flow [22].

In public transportation, AI is used to improve operational efficiency and passenger experience [23]. AI is instrumental in predictive maintenance for transportation fleets. By analyzing data from sensors embedded in vehicles, AI algorithms can predict potential mechanical issues before they escalate. This proactive approach helps transportation companies minimize downtime, reduce maintenance costs, and enhance the overall fleet reliability.

Artificial Intelligence contributes to the development of smart traffic management systems in smart cities. By analyzing historical traffic patterns and current data, AI algorithms can recommend optimal routes for drivers; this not only improves the efficiency of individual journeys but also contributes to the overall sustainability of urban transportation. Machine learning algorithms can analyze historical data to identify patterns and trends, allowing transportation companies to make informed decisions and respond dynamically to changing conditions [24].

#### 2.2.3. Intelligent Transportation Systems (ITSs)

Intelligent Transportation Systems represent a comprehensive framework that applies advanced technologies to various modes of transportation. It encompasses a wide array of applications, including traffic management, public transportation, and vehicle-to-infrastructure communication; its primary objective is to enhance the efficiency, safety, and sustainability of transportation systems [25]. Some of the components of ITSs include real-time traffic monitoring, adaptive traffic signal control, the integration of smart infrastructure, and many more, and these elements work in harmony to not only optimize traffic flow but also reduce travel time and minimize collisions with buildings. In addition, ITSs facilitate the implementation of smart parking solutions, electric vehicle charging infrastructure, and real-time public transportation information systems [26].

A study performed by [27] defines the integration of ITSs and smart cities as emphasizing the role of information technology and addressing urbanizing them to solve transportation challenges. Mobility is a key focus of smart cities; it underlines the development of Intelligent Transportation solutions that greatly reduce congestion, enhance safety, and promote sustainable modes of travel. The integration of advanced technologies, such as the Internet of Things (IoT), Artificial Intelligence (AI), and data analytics, plays a pivotal role in achieving these goals [28].

### 2.3. Sensor-Based Emission Measurement Systems

The rising concern over air pollution and climate change necessitates accurate and efficient methods for monitoring vehicle emissions. Traditional emission measurement systems are often expensive and time-consuming. Sensor-based emission measurement systems offer a promising alternative, enabling the real-time monitoring and assessment of vehicle emissions. A study by [29] developed a low-cost sensor system for measuring CO, NO_2_, CO_2_, PM2.5, and the temperature in motor vehicle exhaust gases. In the study, there was a notable difference between the sensor results and the official results, due to the limited reading range of the low-cost sensor. Research by [30] investigated the use of portable and laboratory instrumentation for measuring gaseous exhaust emissions from light-duty vehicles; it used a low-cost sensor similar to that used in the study by [31], which aimed to determine the exhaust emission values from motor vehicles. The study offered unique insights into the key parameters needed for emission calculations, including pollutant concentrations, exhaust flow, and engine work cycles. The research presented by [32] examines the relationship between traffic patterns and CO_2_ emissions. By leveraging IoT traffic sensors, their model offers a more precise method for estimating emissions in real time, accounting for both congestion and seasonal variations.

The paper by [33] presents the design and implementation of a portable, non-invasive sensor-based CO_2_ emission monitoring system that integrates directly with a vehicle’s Controller Area Network (CAN) bus. For the research, a SprintIR-R CO_2_ Sensor was used. It is a Non-Dispersive Infrared (NDIR) sensor designed to measure CO_2_ concentrations of up to 20%, which makes it suitable for the high concentrations often found in vehicle exhaust emissions. NDIR technology works by measuring the absorption of infrared light by CO_2_ molecules, which provides a reliable and accurate indication of the gas concentration [34].

### 2.4. Machine Learning Models for Calculating CO_2_ Emissions in Cars

There are several machine learning models and programs that have been implemented to ascertain and predict GHG emissions (particularly of CO_2_) in various sectors. One of the most popular programs for calculating GHG emissions is COPERT (Computer Programme to Calculate Emissions from Road Transport). COPERT (v5.7.2) is a computer software developed by the Laboratory of Applied Thermodynamics (LAT) at the Aristotle University of Thessaloniki. The development was funded by the European Environment Agency (EEA), and the majority of GHG emissions in Europe are calculated using COPERT [35]. Unlike most models, COPERT can be used in calculating GHG emissions and air pollutants like NH_3_, SO_2_, CO_2_, N_2_O, VOC, CO, NO_x_, and CH_4_, among others, emitted by passenger cars, large goods vehicles, light goods vehicles, motorcycles, and mopeds [36]. The equation for calculating the emissions from vehicles with the COPERT model is as follows:$$E_{TOTAL}=E_{URBAN}+E_{RURAL}+E_{HIGHWAY}$$
where

*E_URBAN_*—emissions during driving in an urban environment (on local roads).

*E_RURAL_*—emissions during driving in inter-city roads.

*E_HIGHWAY_*—emissions during driving on highways.

The COPERT model has some limitations when calculating the annual CO_2_ emissions from vehicles in a specific country because when calculating the emissions from vehicles, there is a need to consider factors like driver behavior, driving conditions, environmental impact, and more. According to the research in [37], aggressive driving, when compared to regular driving, results in higher emissions. These factors vary significantly based on a country’s historical, social, economic, and cultural context, as well as the general and technical education levels of its population. This led to the development of the REMODIO (Racunanje EMisije Ogljikovega DIOksida) model at the Institute for Energy, Process and Environmental Engineering. This model is tailored to account for the specific factors influencing CO_2_ emissions from vehicles in different regions or countries [38]. The equation for calculating the annual CO_2_ emission from vehicles with the REMODIO model is as follows:$$E_{CO_{2}}=\sum_{g=1}^{2}{\left(N_{r}\cdot l_{r}\cdot g_{r}\cdot K_{z}\right)}_{g\;}\left[\frac{kg\;CO_{2}}{gogini}\right]$$
where

*N_r_*—the number of registered passenger cars at the end of a fiscal year.

*l_r_*—the average annual traversed road of “average vehicle” [km/year].

*g_r_*—the average specific fuel consumption of “an average vehicle” [L/100 km].

*K_z_*—the emission factor, namely, the formation factor CO_2_ [kg CO_2_/L fuel].

*g* = 1—vehicles with a petrol engine.

*g* = 2—vehicles with a diesel engine.

Researchers at the Institute for Energy, Process, and Engineering of the Environment at the Faculty for Mechanical Engineering in Maribor, combined the unique features of the COPERT and REMODIO models to create a new model capable of predicting the decrease in the amount of CO_2_ emitted because of inefficient combustion, ineffective exhaust gas treatment system, and some additional factors not included in earlier models [39]. This model is known as the REPAS model (Računanje Emisije Putničkih Automobila u Saobraćaju), and the software created from this is known as REPAS 1.1. This software was used to calculate the individual CO_2_ emissions and total CO_2_ emissions of vehicles registered in Montenegro in the year 2003 [40]. The current REPAS model is limited to calculating only CO_2_ emissions, and the program has not been updated for some years now. The equation for calculating the annual CO_2_ emission from vehicles with the REPAS model is as follows:
$$E_{{\mathrm{CO}}_{2}}=\sum_{i=1}^{n}[N_{i}\cdot l_{i}(\sum_{j=1}^{m}g_{{\mathrm{ECE}}_{j}}\cdot r_{i})\cdot K_{g_{\mathrm{ECDi}}}\cdot K_{zi}\cdot K_{nsi}][\frac{\mathrm{tons}}{\mathrm{years}}]$$
where

*i*—the number of vehicle categories (*i* = 1, …, 32).

*N_i_*—the number of registered vehicles in the observed category “*i*” [vehicles/year].

*l_i_*—the average annual mileage of category “*i*” [km/vehicle, year].

*j*—the number of types per category “*i*” (*j* = 1, …, *m*).

$g_{{ECE}_{j}}$—manufacturer ECE test specific fuel consumption of PC model “*j*” of category “*i*” [1 fuel/100 km].

*r_j_*—the share of type ”*j*” in category ”*i*”.

$K_{g_{ECDi}}$—the worsening degree of the manufacturer-indicated fuel consumption of vehicle category “*i*”.

*K_zi_*—the emission factor of vehicle category “*i*” [kg CO_2_/liters of fuel].

*K_nsi_*—the incomplete combustion coefficient of vehicle category “*i*”.

Using a traffic simulation model (VISSIM) and an emissions model (MOVES), researchers have predicted the greenhouse gas emitted by vehicles on a limited-access roadway [41]. MOVES (MOtor Vehicle Emission Simulator) is a model created by the U.S. Environmental Protection Agency (EPA) for estimating greenhouse gases, air pollutants, and toxic air emissions from vehicles at the national, county, and project levels [42]. The research by [43] on using a revamped VSP-based MOVES model in Hyderabad, India, revealed that MOVES may not be applicable in countries with bad roads and traffic congestion, because the emission rates calculated in the research were higher than the base rates from MOVES.

CORSIM is a micro-scale model that uses look-up tables to predict HC, CO, and NO_x_ emissions using dynamometer data. CORSIM calculates the total emissions for each link by applying default emission rates to each vehicle for every second the vehicle travels on a specific connection based on its speed and acceleration [44].

The Georgia Institute of Technology created the Mobile Emission Assessment System for Urban and Regional Evaluation to estimate CO, NO_x_, and VOCs [45]. MEASURE is a model that considers the vehicle’s operating modes, including cruising, acceleration, deceleration, and idle status for predicting GHG emissions, instead of estimating exhaust emissions based on the average vehicle speed. The model is data-intensive to use because it includes more than 30 variables and it also does not estimate CO_2_ emissions [46].

Research by [47] on NO_2_ and NO_x_ emissions in vehicles for the Sheffield City Council revealed that diesel vehicles were the top contributors of NO_2_ and NO_x_ emissions in urban Sheffield. The vehicle emissions were estimated using PHEM. PHEM (Passenger Car and Heavy-Duty Emission Model) is a leading Instantaneous Emission Model (IEM) that can simulate fuel consumption as well as the tailpipe emissions of NO_x_, NO_2_, and HCs, particulate mass (PM10), particle number (PN), carbon monoxide (CO), and hydrocarbons (HCs) from an entire fleet of vehicles, ranging from Euro 0 to Euro 6 [48]. This includes heavy-duty vehicles, passenger cars, and light commercial vehicles. PHEM was developed in 2000 as part of the ARTEMIS, COST346, and HBEFA research programs at the Technical University of Graz (TUG, AU). The PHEM model is based on the engine speed–power emission maps of light-duty and heavy-duty vehicles, which were created from engine and chassis dynamometer measurements [47]. See Table 1.

Recently, researchers have been using machine learning and deep learning models for predicting CO_2_ emissions in vehicles. In the research conducted by [59], they trained and tested an artificial neural network (ANN) model, which was used for predicting the combustion and ignition characteristics of diesel–biodiesel–gasoline blend combustion using variables like speed, load, compression ratio, injection temperature, injection pressure, etc., for training the model [59]. ANN was also used in creating Extreme Learning Machine (ELM) models to predict the pressure of a spark ignition single-cylinder engine, which was able to predict the mean effective pressure and was consistent with the experimental results [60]. The researchers at West Virginia University developed ANN models for predicting CO_2_ emissions and NO_x_ emissions in heavy-duty vehicles. The models were trained on the data gathered from transient dynamometer testing of heavy-duty diesel engines, and the predicted CO_2_ emissions and NO_x_ emissions were compared against the real emission data gathered from chassis dynamometer tests of similar vehicles. Although the ANN model did not accurately predict the emissions in lightweight trucks, the prediction accuracy increased as the weight of the trucks increased [61].

An ANN model was used in a study [62] to identify the torque and fuel consumption characteristics of gasoline engines. The brake-specific fuel consumption and engine torque were accurately predicted using an ANN model with a back-propagation learning algorithm. A study by [63] used a similar ANN for predicting a gasoline engine’s performance and exhaust emissions. According to that study, the ANN model can reliably estimate brake-specific fuel consumption, CO emissions, and HC emissions, suggesting that it may be used in place of more traditional modeling methods to predict the Internal Combustion Engine performance and emissions.

In the research conducted by [64], several machine learning models were developed to predict CO_2_ emissions and estimate the fuel consumption of light-duty vehicles. With an accuracy of up to 98.6%, the results revealed that the Univariate Polynomial Regression model is the best model for making predictions based on a single vehicle attribute input. With an accuracy of around 75%, multiple linear regression and multivariate polynomial regression are also useful models for making predictions from various vehicle attribute inputs. The prediction accuracy of the Convolutional Neural Network is over 70%, making it a reliable and promising technology that needs to be further explored [64]. Similarly, a Random Forest regression model was used to predict NO_x_ and CO_2_ emissions after modifying a diesel engine to operate with two types of fuel—diesel and natural gas. Using feature engineering, the RF model was able to accurately predict the NO_x_ emissions [65].

Research by Ağbulut estimated Turkey’s energy consumption and CO_2_ emissions from transportation by using three machine learning algorithms: support vector machine (SVM), artificial neural network (ANN), and deep learning (DL) [66]. The input parameters used to train the algorithms were the population, year, gross domestic product per capita (GDP), and vehicle-kilometer between 1970 and 2016. To compare the predicted outcomes, six statistical metrics were evaluated: mean absolute bias error (MABE), relative root mean square error (rRMSE), mean bias error (MBE), coefficient of determination (R2), Mean Absolute Percentage Error (MAPE), and root mean square error (RMSE). They developed two mathematical models for predicting the energy demand and CO_2_ emissions from transportation, and the results predicted up to the year 2050 [66].

## 3. Methodology

### 3.1. Quantitative and Predictive Analysis

While conducting this research, various datasets were considered, one of which was the dataset on Fuel Consumption Ratings, collected by the Government of Canada for estimating CO_2_ emissions and fuel consumption in light-duty vehicles sold by retailers [67]. This dataset has been used by different researchers in predicting CO_2_ emissions in passenger vehicles. The data are only limited to predicting CO_2_ emission and do not apply to other greenhouse gases and air pollutants like NO_x_, CO, and THC. Another dataset considered is the dataset provided by the European Environment Agency on the CO_2_ emissions from new passenger cars [68]. It is a comprehensive dataset with over 9,000,000 entries, and multiple fields to describe the vehicle. Exploring the EEP data revealed that a large number of rows were duplicated, and the size of the dataframe is not suitable for tools like Jupyter Notebook because of its large size. The third option was the datasets collated by the Vehicle Certification Agency of the United Kingdom for car fuel and emission information. It has data collected from the years 2000 to 2023, although the various datasets used different methods for the measurements of variables over the years. The recent datasets utilized the WLTP for measuring fuel economy and CO_2_ emission [69]. See Figure 2.

### 3.2. Dataset Selection

In this research, to analyze the digital transformation of the transportation sector and its impact on greenhouse gas emissions, the dataset used for the quantitative and predictive analysis was one collated by the Vehicle Certification Agency of the United Kingdom [70]. This dataset contains the fuel consumption, CO_2_ emission, NO_x_ emission, and other tailpipe emission performance figures of new cars currently being sold in the UK. The available version of the dataset is somewhat raw and requires cleaning. The dataset from the years 2020 to 2023 was downloaded and loaded onto Jupyter Notebook using ISO-8859-1 [71]. Jupyter Notebook is an open-source interactive web application that is used as a computational notebook. It is compatible with programming languages like Python (v3.12.0) (Py), R (v4.3.2), and Julia (v1.10.0-rc2) (Ju), and allows researchers to create documents that include software code, computational output, explanatory texts, data visualization, and multimedia files. Although computational notebooks have been around for a while, Jupyter Notebook has become a very popular tool used for data science-related projects [72].

The datasets were loaded on Jupyter Notebook, and the shape of each dataset was checked. The output (Figure A1) revealed that the dataset for 2020 has 4879 rows and 45 columns, the 2021 dataset has 4732 rows and 45 columns, the 2022 dataset has 4727 rows and 45 columns, and the 2023 dataset has 4473 rows and 45 columns.

The datasets were merged to create a unique dataset with 20,644 rows and 47 columns; the reason for the increased number of rows is because of standardization of the column names.

The merged dataset contains some columns with missing values. Here is a summary of the missing values along with their percentages:i.Manufacturer, model, description, etc.: minor missing values, around 0.2%.ii.Transmission: about 3.66% missing.iii.Engine power (Kw/PS): missing values range from about 1.19% to 1.86%.iv.Electric energy consumption Miles/kWh, wh/km, maximum range (Km/Miles), etc.: high percentage of missing values, exceeding 50%.v.WLTP metrics: varies significantly, with some columns having a small percentage of missing values and others having more.vi.Annual costs, noise level, emissions data: varying degrees of missing data, with some columns missing over 50% of their values.

### 3.3. Data Cleaning

The large amount of missing data in certain columns, especially those related to electric energy consumption, ranges, and emissions, is due to the type of components of the vehicles. For instance, electric vehicle-specific metrics like ‘electric energy consumption’ or ‘electric range’ do not apply to Internal Combustion Engine (ICE) vehicles. Similarly, the emissions columns do not apply to vehicles using only electricity as their fuel type. This could result in missing values for these columns in rows representing non-electric vehicles.

There are several methods for handling missing data in a dataset, and one needs to consider the impact of the selected method on the dataset. The research by [73] discusses the advantages and disadvantages of using methods like listwise deletion, mean replacement, and multiple imputation, and the impact it can have on the results. Considering the focus of this research is on greenhouse gas emissions, most of the columns with missing values are not relevant for the prediction of CO_2_ emitted by light-duty vehicles. Dropping some of the attributes like annual costs, electric energy consumption Miles/kWh, etc., would not have a negative impact on the dataset. Also, in the United States and the United Kingdom, fuel consumption is commonly measured in miles per gallon (mpg). But, for evaluating fuel consumption in this dataset, fuel consumption measured in liters per 100 km (L/100 km) serves as a more practical metric, because CO_2_ emission is measured in grams per kilometers (g/km). As fuel economy becomes better, the mpg value rises, while an enhanced fuel efficiency leads to a decrease in liters per 100 km [74].

While the research in [75] discusses both the noise and CO_2_ emissions from vehicles, it does not explicitly establish a direct relationship between the two. Each is influenced by similar factors like vehicle speed and acceleration, but they are assessed independently in terms of their environmental impact. For instance, noise emissions are heavily influenced by acceleration and speed, while CO_2_ emissions are tied to factors like acceleration patterns and the overall vehicle efficiency. Although the focus of the study is primarily on assessing noise and pollutant emissions (including CO_2_) as separate entities, influenced by various factors like vehicle speed, acceleration, and road slope, it does not directly indicate a correlation between noise levels and CO_2_ emissions from vehicles. These findings informed the decision to remove the “noise level dB(A)” column in the dataset.

The engine capacity of a vehicle is measured by the amount of fuel and air that can pass through the cylinders of the measured vehicle and is measured in cubic centimeters (cc). For this research, the engine capacity of the dataset is converted to liters, which is the metric for measuring the size of an engine [76]. After converting the engine capacity to liters, it was approximated to 1 decimal place to obtain the exact engine size of the vehicle.
$$Engine\;Capacity\left(L\right)=\frac{Engine\;Capacity\;(cc)}{1000}$$

After dropping the null values and duplicate rows, the cleaned dataset is exported as a CSV file. This is the dataset that is used for developing/training the machine learning model for predicting the CO_2_ emissions in light-duty vehicles.

### 3.4. Data Exploration and Pre-Processing

#### 3.4.1. Descriptive Categorical and Statistical Analysis

Loading the cleaned dataset and checking its shape showed it has 6629 rows and 9 columns. A categorical analysis of the dataset revealed that there are 40 manufacturers, and MERCEDES-BENZ, FORD, RENAULT, KIA, and SEAT have the highest number of vehicles in the dataset. Also, there are several models produced by each manufacturer. There is a wide range of transmission technologies used by various manufacturers. These include manual transmissions, automatic transmissions, continuously variable transmissions (CVTs), dual-clutch transmissions (DCTs), semi-automatic transmissions (SATs), and All-Wheel Drive Variants. In the dataset, manual transmission is usually denoted by ‘M’ followed by a number (e.g., M5, M6, MT5, MT6). Automatic transmissions are symbolized by ‘A’ followed by a number (e.g., A6, A7, A8, AT). Continuously variable transmissions (CVTs) are represented by ‘CVT’ or variations like ‘E-CVT’. The dual-clutch transmissions (DCTs) can be identified by ‘DCT’ followed by a number (e.g., DCT6, DCT7, DCT8). Similarly, the semi-automatic transmissions are indicated by ‘SAT’ (e.g., SAT5, SAT6), and the All-Wheel Drive Variants are for transmission types with an AWD (All-Wheel Drive) feature, such as ‘A8-AWD’, ‘M6-AWD’, ‘MPS6-AWD’, ‘8A AWD’, ‘8A-AWD’, ‘A6-AWD’, or ‘8AT-AWD’. See Figure 3.

The dataset has 7 categories of fuel types, which are petrol, diesel, electricity/petrol, petrol electric, diesel electric, petrol/LPG (liquefied petroleum gas), and petrol hybrid. The most popular fuel is petrol, which is used by 3207 vehicles. Petrol, which is also known as gasoline or ‘gas’ in North America, is produced from crude oil. Examining the dataset revealed that the most popular powertrain being used by the vehicle is the Internal Combustion Engine (ICE). Internal Combustion Engine (ICE) vehicles use petrol, diesel, compressed natural gas (CNG), or liquefied petroleum gas (LPG) as their fuel type. Vehicles with ICEs tend to emit more CO_2_ from the tailpipe when compared with hybrid electrical vehicles. See Table 2.

#### 3.4.2. Managing Outliers

Outliers are extreme data points in a group of results, significantly different from the other values in the group. Outliers can negatively impact the output received when training a predictive model. It is essential that outliers are checked and removed before analyzing a dataset or training a machine learning model to preventing skewing [77]. See Figure 4 and Figure 5.

#### 3.4.3. Correlation

A heatmap diagram is used to display the correlation coefficient of two variables in the numerical categories of the dataset. Correlation is used to show the relationship between variables. The numerical values range from −1 to +1, with −1 being the maximum negative correlation of the variables, −1 being the maximum positive correlation between the selected features, and 0 showing there is no sign of correlation [78]. See Figure 6.

##### Insights from the Correlation

The correlation between engine power (PS) and fuel consumption comb (L/100 km) is 0.81, indicating that there is a very strong positive correlation between the variables. So, an increase in the engine power of a vehicle will increase the quantity of fuel consumed by the vehicle.The correlation between engine power (PS) and CO_2_ emissions (G/Km) is strong at 0.77, meaning CO_2_ emissions are likely to be low in vehicles with low engine power (PS).There is a 0.87 correlation between engine power (PS) and engine capacity (L), meaning a car with a large engine size will be more powerful than a car with a smaller engine capacity.There is a very high positive correlation (0.94) between fuel consumption comb (L/100 km) and CO_2_ emissions (G/km), meaning the more fuel a vehicle consumes, the more CO_2_ emissions it produces.The relationship between fuel consumption comb (L/100 km) and engine capacity (L) has a correlation coefficient of 0.76, so it is a strong positive relationship.Similarly, the correlation between CO_2_ emissions (G/Km) and engine capacity (L) is a little below average at 0.78. This indicates a strong positive relationship between the variables.

In summary, the engine power (PS), engine capacity (L), and fuel consumption (L/100 km) are features of a vehicle that impact the CO_2_ emissions of vehicles. This answers the third research question.

### 3.5. Building the Machine Learning Model

Artificial Intelligence (AI) refers to the process by which systems or machines imitate human intelligence. This includes features and abilities such as perception, prediction, planning, reasoning, and learning. The field of AI research is rapidly growing due to the interest and contributions of scientists and organizations working towards creating autonomous systems [79]. Some of the key areas of AI research include deep learning (DL) algorithms, knowledge graphs, natural language processing, and machine learning (ML) algorithms [80]. In this research, six regression models were developed, trained, and tested, and the best-performing model was selected for predicting the CO_2_ emissions of light-duty vehicles.

#### 3.5.1. Cleaned Dataset Variables

The dataset’s attributes can be categorized into the following types:

i.Numerical Variables:Target variable: CO_2_ emissions (g/km)—This variable represents the grams of CO_2_ emitted per kilometer driven. It is the variable we aim to predict.Engine power (PS): This variable is directly related to fuel consumption and CO_2_ emissions. Higher power generally implies higher fuel consumption and emissions.Fuel consumption comb (L/100 km): Combined fuel consumption liters per 100 km. This is a highly relevant and strongly correlated predictor of CO_2_ emissions.Engine capacity (L): Engine displacement in liters. A larger engine capacity often, but not always, correlates with higher power and fuel consumption.ii.Categorical Variables:Manufacturer: The brand of the vehicle. This is a high-cardinality categorical variable with numerous unique values (initially 40).Model: The specific model of the vehicle. This is a very high-cardinality categorical variable (initially 656 unique values).Transmission: Describes the type of transmission (e.g., manual, automatic, CVT). It is a moderately high-cardinality categorical variable (initially 54 unique values), with some ambiguity in naming conventions.Fuel type: Indicates the type of fuel used (e.g., petrol, diesel, petrol electric). This is a low-cardinality categorical variable (initially 7 unique values) with significant influence on CO_2_ emissions.Powertrain: Specifies the powertrain configuration (e.g., Internal Combustion Engine (ICE) or Hybrid Electric Vehicle (HEV)). This low-cardinality categorical variable (initially 5 unique values) is a strong predictor, as different powertrain types vary considerably in their fuel efficiency.

#### 3.5.2. Feature Selection Process

The feature selection process was crucial due to the presence of high-cardinality categorical features and the need to balance predictive accuracy with model interpretability and computational efficiency. The steps involved were as follows:i.Initial Feature Elimination: The high-cardinality features manufacturer and model were removed. Including these features would lead to a very high-dimensional feature space after one-hot encoding, increasing computational complexity and significantly increasing the risk of overfitting. The impact of these features may be implicitly captured by other features (e.g., engine type and fuel efficiency). This step is a common practice when dealing with high-cardinality categorical predictors in regression tasks.ii.Pre-processing of Remaining Categorical Features: The remaining categorical features (transmission, fuel type, powertrain) were pre-processed to ensure consistency. This involved replacing spaces and hyphens with underscores (“_”) to standardize the categorical variable labels, preventing issues caused by inconsistent representations. This is standard practice to improve model performance with categorical data.iii.One-Hot Encoding: One-hot encoding was applied to the pre-processed categorical variables to transform them into numerical representations suitable for model training. This is a widely used and effective method for encoding categorical data in machine learning.iv.Standardization of Numerical Features: The continuous features (engine power (PS), fuel consumption comb (L/100 km), engine capacity (L)) were standardized to ensure they have zero mean and unit variance. This helps prevent features with larger scales from dominating the model and improves the performance of some algorithms like linear regression.v.Outlier Removal: Exploratory data analysis using scatter plots revealed outliers, primarily in the relationship between fuel consumption and CO_2_ emissions, indicating potential data errors. These outliers were removed. This step is crucial because outliers can disproportionately affect the performance of some regression models, particularly linear ones.

#### 3.5.3. Regression Analysis

Regression analysis is a method for developing predictive models by analyzing the relationship between a dependent variable and an independent variable [81]. In this research, the target variable is CO_2_ emissions, and the dependent variables are the selected features.

Before training the model, the data were split into a training and testing ratio using the “train_test_split” function from scikit-learn. The test size of the splitting was 0.2, meaning 80% of the dataset was used for training the model, while the remaining 20% was allocated for testing the model.

Several regression models were created and used in predicting the CO_2_ emitted by light-duty vehicles using the United Kingdom VCA dataset.

I.Linear regression: Linear regression is applicable when the dependent variables and the variable to be predicted are in a linear relationship. The most popular types of linear regression are simple linear regression and multiple linear regression [82]. Linear regression models can be quickly trained, are easy to understand, and have a stable performance compared to other algorithms [83].
$$y=w_{1}x_{1}+w_{2}x_{2}\ldots +w_{n}x_{n}+b$$
where *y* is the dependent value to be predicted, x is the independent value, and w is the intercepted values when the value of y and x is zero.

II.Random Forest regression: This is a machine learning algorithm that uses the ensemble method to group and use the decisions from multiple models to accurately make a prediction [84].
$$g\left(x\right)=f_{0}\left(x\right)+f_{1}\left(x\right)+f_{2}\left(x\right)\ldots$$
where *g* is the final predicted model, and $f_{0},\;f_{1},\;f_{2}$ etc., are the base models.

Other algorithms used in creating models during the modeling phase are Decision Tree Regression, Gradient Boosting regression, Lasso Regression, and Ridge Regression.

### 3.6. Performance Parameters

The below performance metrics were used for evaluating the models.

#### 3.6.1. Mean Squared Error

The mean square error (MSE) of a model (of a process for assessing an unobserved variable) is the average of the squares of the errors or the average squared difference between the estimated values and the actual value. MSE is a risk function that represents the expected value of the squared error loss. With the MSE, a model’s performance is assessed.
$$MSE=\frac{1}{N}\sum_{i=1}^{N}{\left(y_{i}-y\right)}^{2}$$
where

*N* = number of forecasts, $y_{i}$ = expected result, and *y* = predicted result.

#### 3.6.2. Root Mean Squared Error

This metric measures the square root of the mean squared error.
$$RMSE=\sqrt{\frac{\sum_{i=1}^{N}{\left(y_{i}-y\right)}^{2}}{N}}$$
where

*N* = number of data points, $y_{i}$ = expected result, and *y* = predicted result.

#### 3.6.3. Mean Absolute Error (MAE)

The Mean Absolute Error is derived by dividing the total absolute error by the sample size in the dataset.
$$MAE=\frac{1}{N}\sum_{i=1}^{N}\left|y_{i}-y\right|$$
where

*N* = number of data points, $y_{i}$ = predicted value of *y*, and *y* = mean value of *y*.

#### 3.6.4. Mean Absolute Percentage Error (MAPE)

Mean Absolute Percentage Error (MAPE) is a statistical measure used to assess the accuracy of a forecasting or prediction model. It calculates the percentage difference between the predicted and actual values, providing a sense of how far off the model’s predictions are on average.
$$MAPE=\frac{1}{N}\sum_{i=1}^{N}\left|\frac{A_{i-}F_{i}}{A_{i}}\right|$$
where

*N* = number of data points, $A_{i}$ = actual value, and $F_{i}$ = predicted value.

## 4. Results and Discussion

The results from Table 3 revealed that the Decision Tree Regression model has the lowest MAE among the various models. With an MAE of 2.20, it is the most accurate prediction of the CO_2_ emissions of the vehicles when compared to the actual values on the dataset. The models with high RMSE scores like the Lasso Regression model might not be able to accurately predict the CO_2_ emissions in the dataset. The models with the lowest RMSE scores are Gradient Boosting (4.57) and Random Forest regression (4.62). The Gradient Boosting and Random Forest models have the highest performance with the MSE metric; this indicates that these ensemble methods are effective for this particular dataset. With an MAPE score of 1.69%, the Decision Tree model is the most accurate model for predicting CO_2_ emissions in vehicles using the features in the dataset. See Figure 7.

Using the performance metrics, it is established that the Decision Tree Regression model is the best for developing the vehicle CO_2_ prediction application. The model is saved using the joblib library in Python; this is performed to preserve all the model has learned and make it possible to be deployed to other environments without having to retrain the model.

### 4.1. Comparative Analysis of the Regression Models

The analysis is based on the Mean Absolute Error (MAE), root mean squared error (RMSE), mean squared error (MSE), and Mean Absolute Percentage Error (MAPE) metrics of the models.

i.Random Forest Regressor:Random Forest is an ensemble method that is less prone to overfitting due to its ability to handle both categorical and numerical features directly.Performance: In this research, the Random Forest Regressor demonstrated excellent performance, achieving a low MAE (2.33), RMSE (6.67), and MAPE (1.81%). This suggests high accuracy and relatively small prediction errors.Strengths: Handles non-linear relationships effectively, is less prone to overfitting than individual Decision Trees, is relatively insensitive to outliers, and can manage high-dimensional data.Weaknesses: The model can be a “black box”, making interpretation difficult.ii.Linear Regression:Linear regression models the relationship between a dependent variable (CO_2_ emissions) and one or more independent variables (features) as a linear equation.Performance: Linear regression exhibited a comparatively higher MAE (3.61), RMSE (7.48), and MAPE (2.73%). This indicates lower accuracy than the ensemble methods.Strengths: Simple to understand and interpret.Weaknesses: Assumes a linear relationship, highly sensitive to outliers, performs poorly with non-linear relationships, and requires careful feature scaling (handled here by StandardScaler during modeling).iii.Decision Tree Regressor:Decision Tree Regression builds a tree-like model where each branch represents a feature, each node represents a decision based on a feature value, and each leaf node represents a predicted value.Performance: Surprisingly, the Decision Tree Regressor achieved the lowest MAE (2.20) and MAPE (1.69%) among all the models. This indicates high accuracy, but the RMSE (7.26) was higher, suggesting some outliers had a bigger impact on the MSE.Strengths: Simple to interpret, handles both categorical and numerical features well, and can capture non-linear relationships.Weaknesses: Highly prone to overfitting, and sensitive to small changes in the training data. The low MAE/MAPE and high RMSE suggest potential overfitting.iv.Gradient Boosting Regressor:Gradient Boosting is another ensemble method that sequentially builds trees, where each subsequent tree corrects the errors made by its predecessors.Performance: Gradient Boosting showed good performance in terms of the MAE (3.13), RMSE (6.37), and MAPE (2.27%). The RMSE was slightly lower than that of Random Forest, suggesting it may be a bit more robust to outliers in this dataset.Strengths: High accuracy, handles non-linear relationships effectively, and less prone to overfitting than a single Decision Tree.Weaknesses: The model is more complex to interpret than linear regression.v.Lasso Regression:Lasso Regression is a linear model that incorporates L1 regularization to shrink coefficients towards zero, effectively performing feature selection and reducing overfitting. The alpha parameter controls the strength of regularization.Performance: Lasso Regression demonstrated the highest MAE (4.54) and RMSE (9.06) and a relatively high MAPE (3.41%). This suggests its performance was inferior to other models, likely due to the alpha value chosen and the non-linear nature of the data.Strengths: Reduces overfitting and performs feature selection.Weaknesses: Sensitive to outliers, assumes a linear relationship. The choice of alpha is crucial and requires careful tuning using techniques such as cross-validation.vi.Ridge Regression:Similarly to Lasso, Ridge Regression is a linear model that uses L2 regularization to constrain coefficient magnitudes. It does not perform feature selection as aggressively as Lasso.Performance: Ridge Regression performed similarly to linear regression, with a relatively high MAE (7.48), RMSE (7.48), and MAPE (2.73%), indicating less predictive power compared to the tree-based and ensemble models.Strengths: Reduces overfitting and handles multicollinearity better than linear regression.Weaknesses: Assumes a linear relationship and does not perform feature selection.

The comparative analysis reveals that tree-based models (Random Forest and Decision Tree) generally outperformed linear models (linear regression, Lasso Regression, and Ridge Regression) while training the models. This is likely because the relationship between the features and CO_2_ emissions is non-linear. While the Decision Tree achieved the lowest MAE and the MAPE indicates its high accuracy, its high RMSE suggests potential overfitting. Gradient Boosting also performed competitively.

### 4.2. Deployment to Streamlit

Streamlit is a Python library for creating interactive web applications for data science and machine learning projects [85]. The trained model was deployed on Streamlit with a user-friendly interface, to enable users to predict and rate the CO_2_ emissions of vehicles by selecting input values on the front end of the web application. Streamlit processes the data entered by the user with the help of the trained model, and the output (CO_2_ emission) is displayed on the front end.

The web application is developed using Python programming language, and Visual Studio Code. After installing the Streamlit library in the vs. code, the application (app.py) was created and coded to enable it to use the trained model that is saved as a joblib file. After developing the app on vs. code, it can be deployed via localhost or online, allowing users to interact with the web app.

## 5. Conclusions

This research revealed the positive impact of digital transformation, and how it integrated emerging technologies to solve problems and improve the way of life of individuals or corporations using different transportation means. It showed how important data are to the digital transformation of the transportation sector and the importance of data analytics and machine learning. Also, it elucidated how Artificial Intelligence can be used in developing efficient, cost-effective, and green solutions for the different modes of transportation. For example, Ultra-Low-Emission Vehicles (ULEVs) and electric vehicles help in reducing the emission of greenhouse gases in our environment, because electric vehicles produce zero (CO_2_) emissions, and the CO_2_ emissions from hybrid vehicles are low when compared with petrol- or diesel fuel-type vehicles.

Sensor-based emission measurement systems are playing an increasingly important role in monitoring and mitigating vehicle emissions. These systems offer advantages in terms of real-time monitoring, cost-effectiveness, and portability. However, addressing challenges related to sensor calibration, environmental factors, and data management is crucial to ensure accurate and reliable emission measurements.

Using a machine learning algorithm and Python programming language, the dataset from the United Kingdom Vehicle Certification Agency (VCA) was used in training the ML model for predicting the CO_2_ emissions of light-duty passenger vehicles. Out of all the models, the Decision Tree Regression model has the lowest MAE. When compared to the actual values on the dataset, it is the most accurate estimate of the vehicles’ CO_2_ emissions, with an MAE of 2.20 and MAPE of 1.69%. The trained model was then deployed as a web application, where users can obtain an estimated CO_2_ emission by inputting values and selecting variables on the web application. This cost-effective solution can be used by car manufacturers, organizations, government agencies, and individuals.

### Limitations and Future Work

This study has potential limitations. The research utilizes data from the UK VCA on light-duty vehicles, and does not consider medium-duty and heavy-duty vehicles, limiting the model’s applicability to other vehicle types. The vehicle’s CO_2_ emissions from the VCA datasets were measured usings WLTP laboratory tests, and external factors such as driver behavior, traffic congestion, or seasonal variations that influence real-world emissions were not considered. Also, this research does not address other greenhouse gases like NO_x_ or THC, which also have significant environmental impacts [86].

Future research will include predictions for other GHGs like NO_x_, THC, and CH_4_ and an exploration of more sophisticated algorithms to improve the prediction accuracy and capture complex relationships among the features of the datasets.

## Figures and Tables

**Figure 1 sensors-24-08219-f001:**
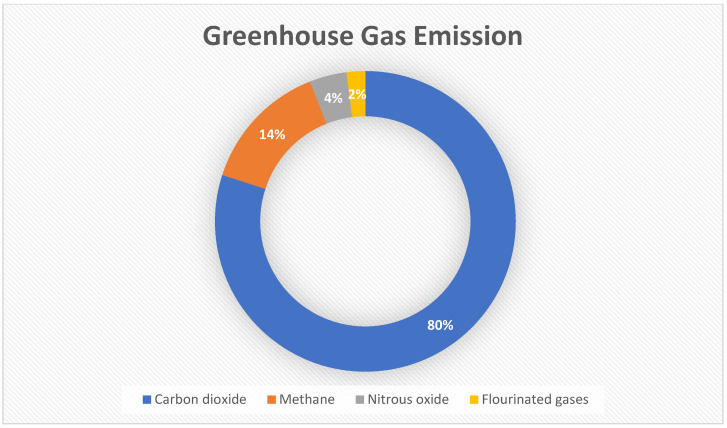
UK Greenhouse gas emission 2022. Source: Department for Energy Security and Net Zero (DESNZ) [3].

**Figure 2 sensors-24-08219-f002:**
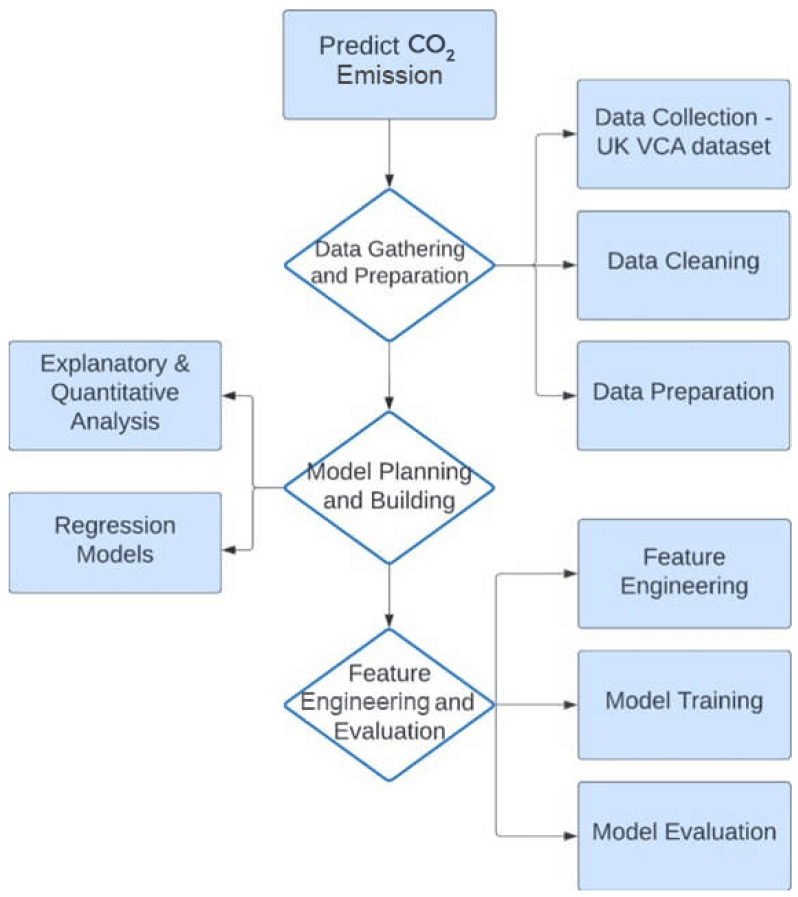
Flow diagram of the predictive analysis.

**Figure 3 sensors-24-08219-f003:**
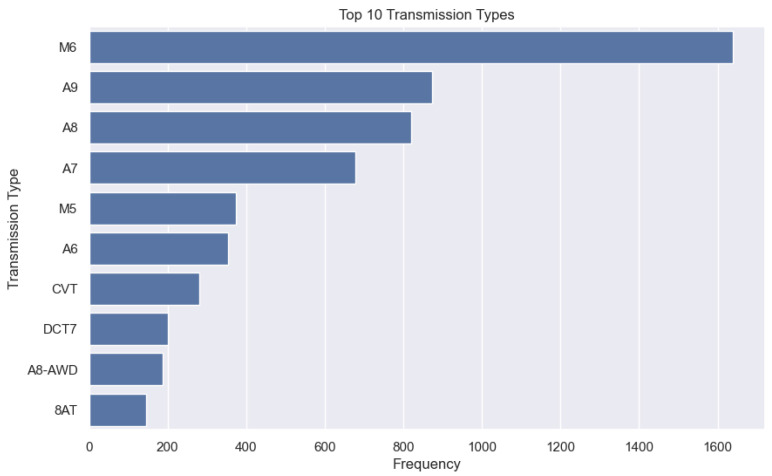
Top 10 transmission types in dataset.

**Figure 4 sensors-24-08219-f004:**
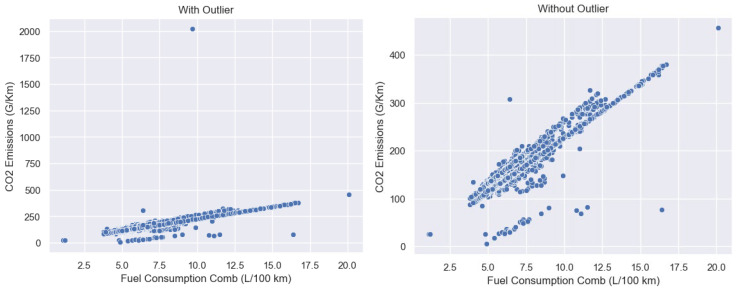
CO_2_ and fuel consumption outlier and without outlier.

**Figure 5 sensors-24-08219-f005:**
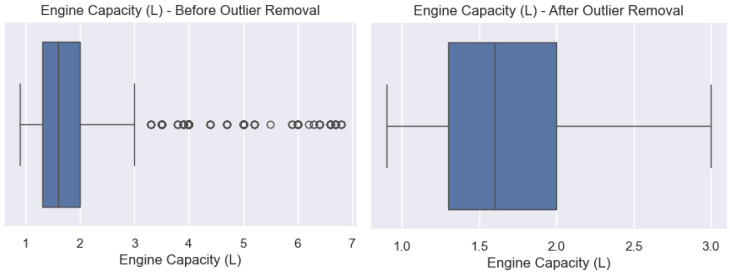
Engine Capacity (L) before and after outlier removal.

**Figure 6 sensors-24-08219-f006:**
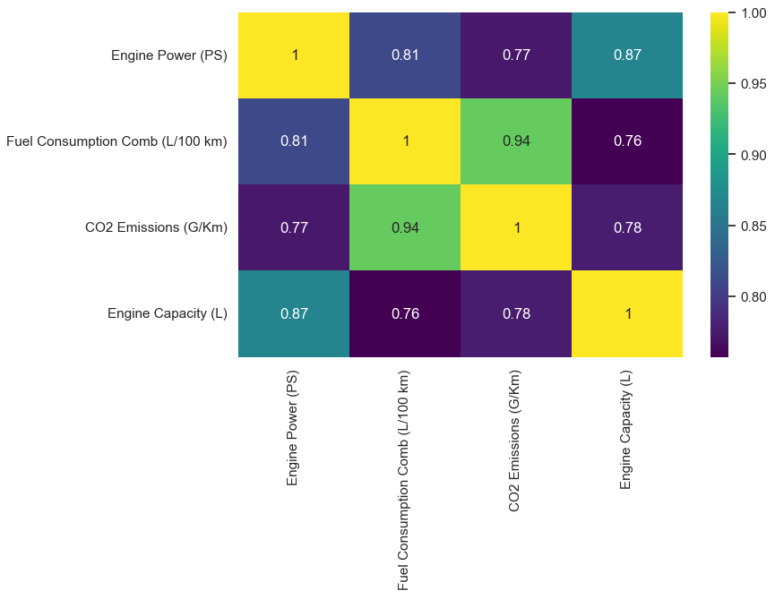
Heatmap correlation between the variables.

**Figure 7 sensors-24-08219-f007:**
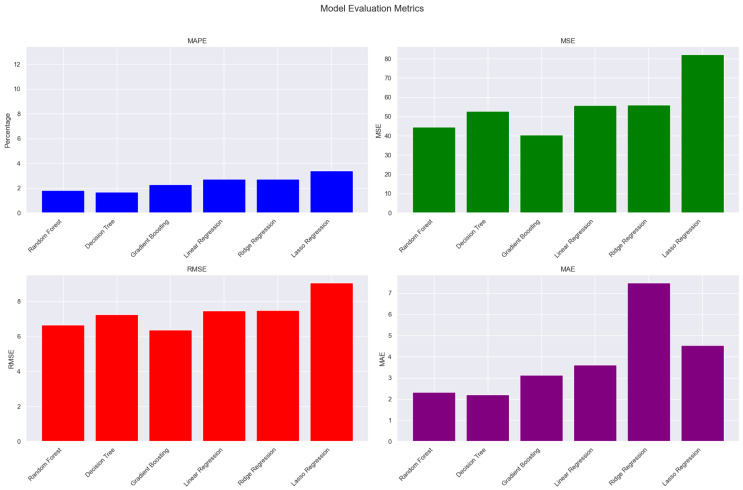
Performance metrics graph.

**Table 1 sensors-24-08219-t001:** Emission models and their input data and features.

Model	Input Data	Features	Source
COPERT	Vehicle category, number of vehicles, weather conditions, load, average speed, distance traveled, etc.	The wide availability of vehicle types and emission components studied.	[49]
MOVES	Vehicle category, number of vehicles, weather conditions, load, average speed, distance traveled, etc.	Ability to calculate emissions for a large number of exhaust components, including HC, CO, CO_2_, NO_x_, CH_4_, N_2_O, and PM.	[50]
PHEM	Among other things, the speed profiles of the vehicles tested.	Accuracy of emission estimation for the entire route, wide range of engine types and test vehicles, time resolution 1 Hz.	[51]
CMEM	Among other things, the speed profiles of the vehicles tested.	In addition to application at the micro scale, it is also possible to estimate emissions at the macro scale, making the model versatile.	[52]
Versit+/Enviver	Speed and acceleration profiles of vehicles.	Automatic generation of emission maps, full support for selected traffic simulation models, e.g., VISSIM.	[53]
VT-Micro	Speed and acceleration profiles of vehicles.	The ability to calculate continuous emissions along the route and fuel consumption for the exhaust gases CO_2_, NO_x_, CO, and THC.	[54]
ESTM BOSH	Speed and acceleration profiles of vehicles.	The possibility of creating emission maps within the scope of the VISSIM (v2024.00-03) software, which allows very precise localization of areas of increased concentrations of exhaust constituents.	[55]
EMPA	Speed and acceleration profiles of vehicles.	Possibility to calculate emissions for LDVs only.	[56]
EMFAC	E.g., average vehicle speed, structure type of vehicles, vehicle load, ambient conditions: temperature, humidity, etc.	Ability to calculate emissions for a number of indicators: THC, CO, NO_x_, PM, SO_x_, and CO_2_.	[57]
MODEM	Speed and acceleration profiles of vehicles.	Continuous emission estimation; no emission estimation possible for heavy-duty vehicles.	[58]

**Table 2 sensors-24-08219-t002:** Descriptive statistics of the numeric variables.

	Count	Mean	Std	Min	25%	50%	75%	Max
Engine power (PS)	6629.00	191.58	115.72	60.00	120.00	150.00	204.00	835.00
Fuel consumption comb (L/100 km)	6629.00	6.97	1.97	1.10	5.70	6.50	7.60	20.10
CO_2_ emissions (G/Km)	6629.00	164.70	51.48	5.00	134.00	153.00	185.00	2019.00
Engine capacity (L)	6629.00	1.86	0.83	0.90	1.30	1.60	2.00	6.80

**Table 3 sensors-24-08219-t003:** Performance parameter results.

	Mean Absolute Error (MAE)	Root Mean Squared Error (RMSE)	Mean Squared Error (MSE)	MAPE
Random Forest Regression	2.33	6.67	44.47	1.81%
Linear Regression	3.61	7.48	55.90	2.73%
Decision Tree Regression	2.20	7.26	52.71	1.69%
Gradient Boosting	3.13	4.57	40.52	2.27%
Lasso Regression	4.54	9.06	82.10	3.41%
Ridge Regression	7.48	7.48	56.00	2.73%

## Data Availability

Data are contained within the article.

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
