# Peer review of "Application of Machine Learning to Predict CO2 Emissions in Light-Duty Vehicles"

_sensors, 2024, doi:10.3390/s24248219_

Round 1
Reviewer 1 Report
Comments and Suggestions for Authors
In this paper, the Regression Machine learning models were developed to predict CO2 emissions. Research shows that models with Ensemble feature like Random Forest and Decision tress performs better with the VCA dataset. However, there are some issues that need to be addressed before acceptance.
Major comments:
(1) Clarity and Structure: The paper is well-organized, but the introduction could benefit from a clearer statement of objectives and contributions.
(2) Methodology: While the use of ensemble models is appropriate, more detailed descriptions of the dataset and feature selection process would improve transparency.
(3) Technical Depth: The technical explanations are sound, but including more comparative analysis with other models could strengthen the discussion.
(4) Limitations and Future Work: Address potential limitations of the study and suggest directions for future research.
Minor comments:
Lines9-19: The abstract needs reorganization to better summarize the paper's accomplishments. Consider adding numerical or quantitative descriptions.
Lines5-8: Please complete the author information in lines 5-8, including affiliations and email addresses.
Introduction (Lines 24-56): More references need to be added. In addition, Section 1.2 is on line 57, so where is section 1.1?
Figures 3-5:The code in Figures 3-5 can be placed in the supplementary materials or appendix, not in the main text. The main text should show more visual results.
The code in Figures 3-5 can be placed in the supplementary materials or appendix, not in the main text. The main text should show more visual results.
Table 3-4: The results throughout the text should retain 1-2 decimal places for consistency.

The English could be improved to more clearly express the research.
Author Response
1. Clarity and Structure: The paper is well-organized, but the introduction could benefit from a clearer statement of objectives and contributions.
Response: Thank you for pointing this out. We agree with this comment. Therefore, we have revised the introduction. Pages 2-3 (Lines 28-50). Also, clearer objectives were added - Page 3 (Lines 52-62).
2. Methodology: While the use of ensemble models is appropriate, more detailed descriptions of the dataset and feature selection process would improve transparency.
Response: Thank you for your feedback. Paper was revised, the cleaned dataset description and feature selection process are on pages 17-19 (Lines 614-681).
3. Technical Depth: The technical explanations are sound, but including more comparative analysis with other models could strengthen the discussion.
Response: Thank you for your feedback. Comparative analysis was added, and is on Pages 21-24 (Lines 775-843)
4. Limitations and Future Work: Address potential limitations of the study and suggest directions for future research.
Response: Thank you for pointing this out. We agree with this comment. Therefore, we have added Limitations and Future work. Pages 26 (Lines 896-906).
Lines9-19: The abstract needs reorganization to better summarize the paper's accomplishments. Consider adding numerical or quantitative descriptions.
Response: Done. Page 1 (Line 9)
Lines5-8: Please complete the author information in lines 5-8, including affiliations and email addresses.
Response: Thank you for pointing this out. Authors information have been added. Page 1 (Lines 5-8)
Introduction (Lines 24-56): More references need to be added. In addition, Section 1.2 is on line 57, so where is section 1.1?
Response: Section 1 has been corrected. And references added to the introduction. Pages 2-3 (Lines 28-110)
Figures 3-5:The code in Figures 3-5 can be placed in the supplementary materials or appendix, not in the main text. The main text should show more visual results.
Response: The screenshots have been removed, and added to appendix. Page 27 (Lines 911-918)
Table 3-4: The results throughout the text should retain 1-2 decimal places for consistency.
Response: Thank you for the feedback, the results have been revised to 2 decimal places. Page 15 (Line 547) and Page 20 (Line 751)
Reviewer 2 Report
Comments and Suggestions for Authors
Despite a good attempt to apply a machine learning technique to air pollution area, some shortcomings are as follows:
1. The abstract contains repetitive background information with no focus on the problem or the contribution of this work. Also the results are not mentioned adequately. This term "Light Vehicles Test Procedure (WLTP)" is defined twice in the abstract. Please rewrite the abstract considering these issues.
2. The introduction lacks references. Please add references for all background information. Also, eliminate the basic and repetitive information in the introduction. Make it more concise focusing on the important information.
3. No need to describe all these methods in details (IOT, AI, ITS), mentioning some examples of their application in the transportation sector could provide better information than the extensive explanation as in the manuscript.
4. Need to revise the numbering of the introduction section and its subsections. Also section 3.4 numbering is not correct.
5. Figures should be referred to in the corresponding location in text.
6. Abbreviations are defined more than one time. Make sure to define them in the first use and use the abbreviation afterwards.
7. The sentence should not start with the number of the cited reference as in line 344 ([36] estimated GHG emissions on a limited-access highway), please revise writing.
8. The introduction is more or less tedious to read due to having extensive background information which is not necessary. And the contribution of this work is not clear.
9. No need to add screenshots from the code in the manuscript, specifically for the basic tasks such as data loading and reading
10. How data was split for training and testing...
Author Response
1. The abstract contains repetitive background information with no focus on the problem or the contribution of this work. Also the results are not mentioned adequately. This term "Light Vehicles Test Procedure (WLTP)" is defined twice in the abstract. Please rewrite the abstract considering these issues.
Response: Thank you for pointing this out. Abstract was revised. Page 1 (Lines 9-24)
2. The introduction lacks references. Please add references for all background information. Also, eliminate the basic and repetitive information in the introduction. Make it more concise focusing on the important information.
Response: Thank you for pointing this out. We agree with this comment. Therefore, we have revised the introduction. Pages 2-3 (Lines 28-50).
3. No need to describe all these methods in details (IOT, AI, ITS), mentioning some examples of their application in the transportation sector could provide better information than the extensive explanation as in the manuscript.
Response: Thank you for your feedback. We revised the section. Pages 4-6 (Lines 149 - 226)
4. Need to revise the numbering of the introduction section and its subsections. Also section 3.4 numbering is not correct.
Response: Thank you for pointing this out. The numbering for section 1 and section 3.4 have been corrected. Pages 2-3 (Lines 28-110), and pages 14-17 (Lines 514-601)
5. Figures should be referred to in the corresponding location in text.
Response: Please clarify
6. Abbreviations are defined more than one time. Make sure to define them in the first use and use the abbreviation afterwards.
Response: WLTP Abbreviations updated
7. The sentence should not start with the number of the cited reference as in line 344 ([36] estimated GHG emissions on a limited-access highway), please revise writing.
Response: Thank you for pointing this out. We agree with is comment, therefore we have revised the sentence. Page 8 (Lines 326-328)
8. The introduction is more or less tedious to read due to having extensive background information which is not necessary. And the contribution of this work is not clear.
Response: Thank you for pointing this out. We agree with this comment. Therefore, we have revised the introduction. Pages 2-3 (Lines 28-50).
9. No need to add screenshots from the code in the manuscript, specifically for the basic tasks such as data loading and reading.
Response: The screenshots have been removed, and added to appendix. Page 27 (Lines 911-918)
10. How data was split for training and testing.
Response: Thank you for your feedback. The data was split 80-20 for training and testing. Page 19 (Lines 688 - 691).
Round 2
Reviewer 1 Report
Comments and Suggestions for Authors
1、Please add the authors' affiliations for Lines 5-8 .
2、In the text, "CO2" should be written as "CO₂" with the "2" as a subscript.
Author Response
1、Please add the authors' affiliations for Lines 5-8.
Response: Thank you for your feedback. The authors' affiliation has been added on pages 17-19 (Lines 5-8).
2、In the text, "CO2" should be written as "CO₂" with the "2" as a subscript.
Response: Thank you for pointing this out. We agree with this comment and have revised the manuscript.
Reviewer 2 Report
Comments and Suggestions for Authors
In order not to be tedious for readers, the volume of the manuscript should be reduced by removing general theoretical background. Table 2 includes too detailed terms glossary. Fig 6 looks enough by stating in the main context. Fig 10 should be clearer. Are Fig 11 and 11 really necessary here?
Author Response
In order not to be tedious for readers, the volume of the manuscript should be reduced by removing general theoretical background. Table 2 includes too detailed terms glossary. Fig 6 looks enough by stating in the main context. Fig 10 should be clearer. Are Fig 11 and 11 really necessary here.
Response: Thank you for your feedback, the manuscript has been revised. Table 2, now TableA1 has been moved to the Appendix (Page 26, Line 930). Also Figure 11 & Figure12, now Figure A3 & Figure A4 were moved to the Appendix. The size of Figure 10, now Figure 7 was slightly increased, to make it clearer.